# Salivary Antibody Response of COVID-19 in Vaccinated and Unvaccinated Young Adult Populations

**DOI:** 10.3390/vaccines10111819

**Published:** 2022-10-28

**Authors:** Sandhya Sundar, Ramya Ramadoss, Rajeshkumar Shanmugham, Lakshmi Trivandrum Anandapadmanabhan, Suganya Paneerselvam, Pratibha Ramani, Rumesa Batul, Mohmed Isaqali Karobari

**Affiliations:** 1Department of Oral Pathology, Saveetha Dental College and Hospitals, Saveetha Institute of Medical and Technical Sciences University, Chennai 600077, Tamil Nadu, India; 2Department of Pharmacology, Saveetha Dental College and Hospitals, Saveetha Institute of Medical and Technical Sciences University, Chennai 600077, Tamil Nadu, India; 3Conservative Dentistry Unit, School of Dental Sciences, Universiti Sains Malaysia, Health Campus, Kubang Kerian 16150, Kelantan, Malaysia; 4Department of Conservative Dentistry and Endodontics, Saveetha Dental College and Hospitals, Saveetha Institute of Medical and Technical Sciences University, Chennai 600077, Tamil Nadu, India

**Keywords:** antibody response, COVID-19, SARS-CoV-2, vaccination, saliva

## Abstract

COVID-19 is a terrible pandemic sweeping the whole world with more than 600 million confirmed cases and 6 million recorded deaths. Vaccination was identified as the sole option that could help in combatting the disease. In this study, SARS-CoV-2 antibodies were assessed in the saliva of vaccinated participants (Covaxin and Covishield) through enzyme-linked sorbent assay (ELISA). The IgG antibody titres in females were significantly greater than those of males. The total antibody titres of vaccinated individuals were greater than those of unvaccinated participants, although not statistically significant. Individuals who had completed both doses of vaccination had higher antibody levels than those who had received a single dose. People who had experienced COVID-19 after vaccination had better immunity compared to those who were unvaccinated with COVID-19 history. Thus, SARS-CoV-2 spike-specific antibodies were successfully demonstrated in saliva samples, and knowledge about the immunity triggered by the vaccines can assist in making informed choices.

## 1. Introduction

The COVID-19 pandemic has caused widespread loss of human lives in most highly populated zones. Mortality rates globally account for more than >5.4 million deaths [1]. Despite the successful launch of vaccines throughout the world, COVID-19 still remains a significant threat to humanity. COVID-19 immunization does not prevent the vaccinated population from contracting the disease, particularly a few months after vaccine delivery, questioning the effectiveness of the available vaccines against COVID-19 and its variants. There are also looming threats regarding the multiple variants of the virus [2]. Considering such a bleak scenario, there is a dire need to know the immune status of the individual.

COVID-19 antibody testing is routinely conducted using serum, and large-scale population-based studies reported on it [3,4,5,6]. However, serum-based testing is invasive and difficult to perform in large-scale population-based studies. It is also challenging in vulnerable groups where obtaining a serum sample may not be feasible. Furthermore, the collection of samples at different time periods for the progressive evaluation of herd immunity using serum samples imposes a heavy burden in terms of feasibility and incurred costs [7].

Saliva is a promising sample source in disease diagnosis and prognosis. It provides a huge advantage over serum by being a non-invasive method. The cooperation of participants in times of repeated sampling is ensured, rendering it a preferred method in large-scale studies. Saliva samples are also less cumbersome in terms of sample storage and collection equipment [8,9]. Salivary antibodies are the highest order of persistent mucosal defence against SARS-CoV-2. Their temporal ambulation in saliva is consistent with those in serum. Salivary IgG responses produce 100% accurate diagnosis [7].

The current study analyses the antibody response (both total Ab and IgG) in the saliva of young vaccinated adults.

## 2. Materials and Methods

The institutional ethical committee of Saveetha Institute of Medical and Technical Sciences, Chennai (IHEC/SDC/FACULTY/21/OPATH/196) approved the study. Study participants gave their informed consent for the collection and use of salivary samples. A total of 88 subjects from a young adult population (18–21 years) were selected. The participants were divided into Group I for vaccinated individuals and Group II for unvaccinated individuals. Clinical information regarding the past history of COVID-19 infection and vaccination was collected appropriately. The proforma of each patient were then reviewed.

### 2.1. Sample Collection

Saliva samples were obtained between the hours of 9 and 11 a.m. under nonstimulatory circumstances. At least one hour prior to collection, participants were requested to refrain from eating, chewing, or drinking. Before saliva collection, subjects were instructed to rinse their mouths with water for at least 1 min. Saliva samples (2 × 1 mL tubes) were collected using a sterile polypropylene container with a wide opening (Tarson 510010 polypropylene/HDPE 100 mL Sample Container Sterile) over a period of about 5 min. Subjects were asked to swallow first, then bend their heads forward and expectorate all of their saliva into the centrifugal tubes for 10 min without swallowing.

Salivary samples were prepared in accordance with a previous study [10]: centrifuged at 1500 rpm for 10 min at 4 °C and concentrated by further repeating the process for 8 and 5 min. The final supernatant was divided into 1 mL aliquots and kept at −80 °C for analysis. Each sample could only go through one freeze–thaw cycle. Salivary samples were stored at −20 °C until analysis.

### 2.2. Elisa Procedure

The reagents were kept at room temperature for at least 30 min before use. Each kit had positive and negative controls to validate the results.

#### 2.2.1. ELISA for Detecting Total Antibody Number

The assay was performed in accordance with the manufacturer’s guidelines. The reagents and microwell plates used for the assay were provided in the Platelia SARS-CoV-2 Total Ab kit (Bio-Rad Laboratories, Inc., Hercules, CA, USA). First, 15 µL of saliva was admixed with 75 µL of conjugate (recombinant SARS nucleocapsid protein associated with horseradish peroxidase) and 60 µL of sample diluent (TRIS-Nacl buffer) in prereaction microwells to produce the sample.

After aspirating and rejecting the sample once, 100 µL of the calibrator, positive and negative controls, and samples were placed to the wells of the reaction microplate and incubated for 30 min at 37 °C. The plates were washed with a microplate washer five times. The substrate buffer (citric acid and sodium acetate solution) and chromogen (TMB solution) were combined in equal proportions; then, 200 µL of the mixed solution was distributed into each well and incubated in the dark for 30 min at room temperature. After that, 100 µL of the stopping solution (H2SO4 1N) was added to each well and carefully mixed to halt the reaction. Within 30 min of administering the stopping solution, the optical density was measured at 450 nm with a reference filter at 630 nm using an ELISA reader (BeneSphera, Faridabad, India). Output reports were produced by subtracting the optical density (OD) at 450 nm from the OD at 630 nm. Following the manufacturer’s advice, data were evaluated, and the results are reported as a ratio. On the basis of the sample OD divided by the average OD of the calibrators, the sample absorbance ratio was calculated.

#### 2.2.2. ELISA for Detecting IgG Immunoglobulin

The assay was performed in accordance with the manufacturer’s guidelines. The reagents and microwell plates used for the assay were available in the Euroimmun Anti-SARS-CoV-2 IgG kit (EUROIMMUN Medizinische Labordiagnostika AG, Waltham, MA, USA). The sample was produced by diluting saliva with sample buffer in a 1:1 ratio, mixing it once, transferring 100 µL of the calibrator, positive and negative controls, and diluted samples into the reaction microplate’s wells, and incubating them for 60 min at 37 °C.

The enzyme conjugate (peroxidase-labelled antihuman IgG—100 µL per well) was applied to each well and incubated at 37 °C for 30 min before being rinsed three times. The chromogen/substrate solution (100 µL per well) was added and incubated at room temperature for 30 min in the dark. The stopping solution (100 µL per well) was added in the same order and the same speed as those with which chromogen/substrate solution was introduced. The optical density was measured at 450 nm with a reference filter at 620 nm on an ELISA reader (BeneSphera, Faridabad, India) within 30 min of adding the stopping solution. The optical density (OD) at 450 nm was subtracted from the OD at 630 nm to create output reports. Data were analysed in accordance with the manufacturer’s recommendations, and the outcomes are presented as a ratio. The sample absorbance ratio was obtained by dividing the sample OD by the average OD of the calibrators.

### 2.3. Statistical Analysis

The tests of normalcy variables did not follow normal distribution according to the results of the Kolmogorov–Smirnov and Shapiro–Wilks tests. As a result, a nonparametric approach was used to analyse the data. The Mann–Whitney U test was used to compare the values of two groups. SPSS (IBM SPSS Statistics for Windows, Version 26.0, Armonk, NY, USA: IBM Corp., Released 2019) was used to analyse the data. The level of significance was set at 5% (=0.05).

## 3. Results

A total of 88 participants were enrolled in the study; 28% (*n* = 25) of them were males and 72% (*n* = 65) were females. All of the participants were in the age group of 18–21 years. The study was conducted in Chennai, India. Most of the participants had Indian nationality and some of the participants were from the UAE. All of the participants had received either the Covishield or Covaxin vaccine (these two vaccines were the only available vaccines in India at the point of conducting the study). Of the individuals, 95.5% (*n* = 85) were in Group I (vaccinated), and 4.5% (*n* = 4) were in Group II (unvaccinated). Of the participants, 53.6% (*n* = 44) had been inoculated with both doses of a COVID-19 vaccine, whereas 46.4% (*n* = 39) had received only the first dose of vaccination.

In addition, 15.9% (*n* = 14) of the study subjects had a COVID-19 history, of which 21.4% (*n* = 3) had experienced the infectious disease before vaccination, and 78.6% (*n* = 11) after vaccination, of which 27.3% (*n* = 3) after the first dose, and 72.7% (*n* = 8) after the second dose. Participants who had had COVID-19 after vaccination acquired it after 2 months +/− 1 month after vaccination. According to the clinical data of the research participants, immunisation prevented hospitalisation, although it was highly common prior to vaccination.

### 3.1. Antibody Titres in Males versus Females

Salivary samples collected from the participants were subjected to two different analyses, total antibody and IgG assessment (through ELISA), and the readings were taken at an optical density of 430 nm. The mean values of IgG and total Ab obtained at 430 nm were 1.03 and 0.75 U/mL, respectively. The mean IgG titres in males and females were 0.93 ± 0.43 and 1.07 ± 0.39, respectively. The total antibody titres in the male and female populations in the study were 0.72 ± 0.24 and 0.76 ± 0.24 mg, respectively (Figure 1).

### 3.2. Antibody Titres in Vaccinated versus Unvaccinated

IgG titres in the vaccinated and unvaccinated populations in the study were 1.02 ± 0.41 and 1.09 ± 0.39 mg, respectively. The total antibody titres in the vaccinated and unvaccinated populations in the study were 0.75 ± 0.24 and 0.73 ± 0.27 mg, respectively (Figure 2). IgG levels were slightly higher in the unvaccinated population, and total antibody levels were greater in vaccinated individuals. The differences in the antibody titres between the vaccinated and unvaccinated populations were not statistically significant (*p*-values = 0.65 and 0.92, respectively).

### 3.3. Antibody-Titre Difference between Single- and Double-Dose-Vaccinated

The IgG titres in single- and double-dose-vaccinated populations were 0.99 ± 0.37 and 1.06 ± 0.44 mg, respectively. The total antibody titres in single- and double-dose-vaccinated populations in the study were 0.73 ± 0.22 and 0.76 ± 0.26 mg, respectively (Figure 3). IgG and total antibody levels were greater in double-dose-vaccinated individuals compared to those of the single-dose-vaccinated population. The differences in the antibody titres between the single- and double-dose-vaccinated populations were not statistically significant (*p*-values = 0.75 and 0.65, respectively).

### 3.4. Antibody-Titre Difference between Vaccinated and Unvaccinated Individuals with COVID-19 History

The IgG in COVID-19-infected individuals with and without vaccination was 1.08 ± 0.49 and 1.03 ± 0.45 mg, respectively. Total antibody titres in the COVID-19-infected population with and without vaccination were 0.78 ± 0.32 and 0.67 ± 0.29 mg, respectively (Figure 4). IgG and total antibody levels in vaccinated individuals with a COVID-19 history were greater compared to those of unvaccinated individuals with a COVID-19 history. IgG and total antibody titre differences in COVID-19-infected individuals with and without vaccination were not statistically significant (*p*-values = 0.81 and 0.48, respectively).

## 4. Discussion

With more than 600 million confirmed cases of COVID-19 and 6 million associated recorded deaths, the world is facing a huge challenge in combating the global pandemic. With more than 7 billion administered COVID-19 vaccine shots, vaccines can add to the armour of first-line defence against this overwhelming struggle. A vast number of COVID-19 vaccines are available in the market that are being developed with different methodologies.

At the time of conducting the study, only Covaxin (BBV152, Bharath Biotech Ltd., India) and Covishield (ChAdOx, Serum Institute of India Ltd., India) were the only COVID-19 vaccines available in India. These are inactivated-viral-particle and recombinant-adenovirus-mediated vaccines, respectively. Both of them target the S protein of SARS-CoV-2 and are given as a two-dose regimen separated by a period of 4 weeks. The World Health Organisation and Central Drugs Standard Control Organisation (CDSCO) approved the usage of these indigenous vaccines to cater for the emergency needs of the 300 million inhabitants of India. Studies have evidenced the adequate trigger of cell-mediated and humoral immunity by these vaccines. The reported efficacy of Covaxin against COVID-19 is 80%, and that of Covishield is 90% [11].

Concerns related to the safety and efficacy of the available vaccines are foremost. The ability of the developed vaccines to provide cross-immunity has been greatly researched, as the receptor-binding domain (RBD) of the virus is subjected to various mutations. Antibody responses triggered by the vaccine are the primary way of measuring their potency and differ between individuals.

Salivary antibodies, as a source of mucosal immunity estimation, should not be overlooked as they are an immediate defence against COVID-19 infection. Saliva, as an effective alternative to nasopharyngeal swabs, has gained popularity in COVID-19 diagnosis in a meta-analysis with a sensitivity of 91% [9]. The presence of SARS-CoV-2 antibodies in the saliva is sparsely referred to in the literature [7,12,13]. SARS-CoV-2-specific mucosal antibodies were even demonstrated in the saliva of children (large paediatric cohort study) [14].

In the present study, IgG and total antibodies were demonstrated in the salivary samples of the participants. Antibody levels in females were significantly higher compared to those of males, which is in accordance with many reported studies [15]. The exaggerated immune response in females can be attributed to the oestrogen sex hormone, which influences the presence of long-lasting circulating antibodies and T lymphocytes compared to that in males [16,17]. The X chromosome encodes many immune-related genes, including CD40L, TLR7, which is virus-sensitive and thereby triggers the production of high levels of Type I interferons [18]. The significant time-dependent reduction in neutralising antibodies (developed after the infection) is also rapid in males compared to females [19], which questions the protection durability of vaccines across the sexes.

The IgG antibody titres of vaccinated participants were not significantly different from those of the unvaccinated. The total antibody responses of vaccinated individuals were slightly greater than those in unvaccinated individuals. Three out of four unvaccinated individuals had previously been infected with COVID-19. A recent COVID-19 infection (2–3 months of duration) could be the reason for the robust production of antibodies in these participants, equalling them to those of vaccinated individuals. IgG, as the antibody that is in play within 10 days of infection and persisting even after 3 months, can be increased in people with previous COVID-19 infection. In addition, the role of herd immunity could be a possible explanation.

The “antibody wear out” occurring a few months after vaccination (antibody kinetics) should be considered to be important, as was reported in many studies [20,21]. Recently, a study by ICMR on 614 fully vaccinated healthcare workers in Bhubaneswar inferred a decline in antibodies 2 months after Covaxin and 4 months after Covishield. Immediately after vaccination, the immune system is targeted to produce an increasing number of antibodies and cytotoxic T lymphocytes along with memory B cells. The protective immunity offered by the antibodies is short-lived, whereas that of cytotoxic and memory cells is long-lasting [22,23].

In this study, the participants demonstrated a good antibody response in salivary samples with high IgG levels compared to total antibodies. Previous studies measuring salivary antibody titres also reported frequently high IgG levels that were stable and persistent even after mild infection up to period of 3 months compared to short-lived salivary IgA and IgM antibodies against COVID-19 [24]. In addition, salivary IgG demonstrated resistance to temperature and treatment with chemicals [25]. Thus, the assessment of IgG in salivary samples is a reliable sensitive method.

The ability of subsequent doses of vaccination in accentuating the immune response has not been much explored. In the present study, the antibody response was increased in participants who had completed both vaccination doses compared to those who were partially vaccinated (first dose only). A robust production of antibodies after the second dose of vaccination was recorded in previous studies [26,27]. An already conditioned/primed immune system escalates the antibody response after a subsequent dose of vaccination. The study by Lapi et al. [28] reinforced the point that two doses of vaccination are a requirement for seronegative (with no prior exposure to COVID-19 antigens) and seropositive (exposed but with no clinical symptoms) individuals to achieve an equal immune response to those who had had a COVID-19 infection.

The occurrence of COVID-19 infection in vaccinated individuals has further challenged the value and efficacy of vaccines. The present study clearly shows that the antibody response in vaccinated individuals with a COVID-19 history was greater than that in those who had experienced COVID-19 infection in an unvaccinated state. This supports the valuable role of the immune response aroused by vaccines in effectively combating the disease without serious morbidities.

## 5. Conclusions

In the present study, vaccinated individuals presented a comparable increase in immune status as that of those who had been infected with SARS-CoV-2. The dynamics of antibody kinetics on the individual level disputes the ‘correlates of protection’ offered by these antibodies. The recent outbreak of the Omicron mutation has challenged the future immunity conferred by vaccines, and there is a need for additional immunisation [29]. Does all this suggest an imperative need for a change in the overall scheme of immunisation?

## 6. Limitations and Future Scope

Serum analysis was not paired with the salivary antibody estimation for a comparative value. The assessment of antibodies in the saliva of currently affected individuals with SARS-CoV-2 infection should give the larger picture. The functional tests of neutralising antibody estimation can indicate the actual state of cellular immunity and response. A future large-scale follow up of the study with larger samples and groups is merited.

## Figures and Tables

**Figure 1 vaccines-10-01819-f001:**
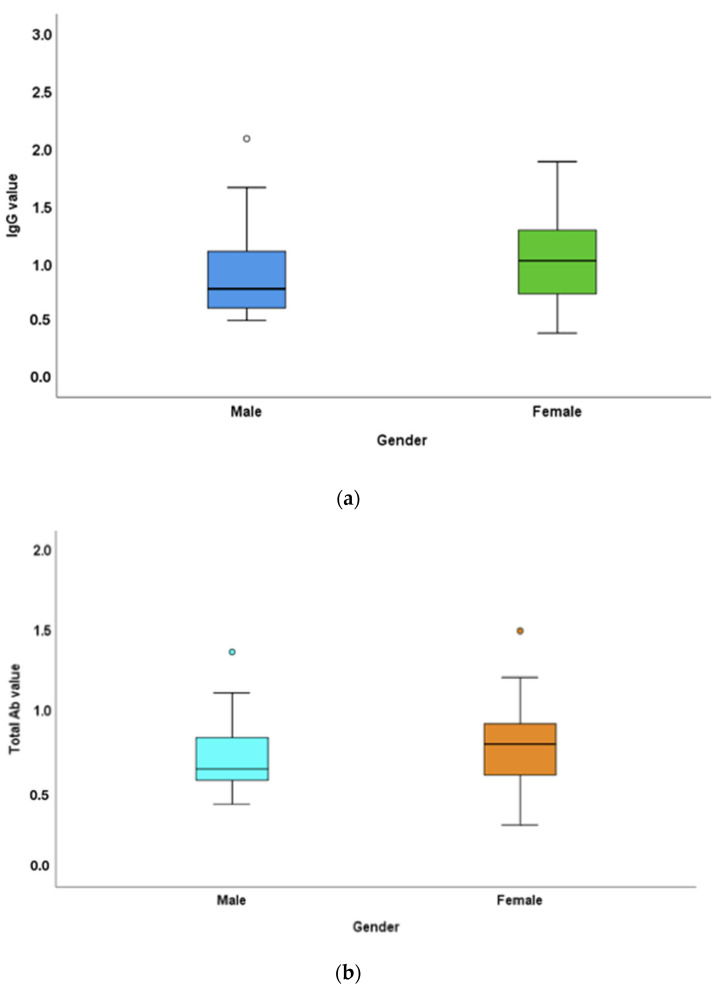
Mean IgG and total antibody titre difference between the male ande female populations in the study: (**a**) IgG titres in males and females in the study were 0.93 ± 0.43 and 1.07 ± 0.39 μg, respectively. (**b**) Total antibody titres in the male and female populations in the study were 0.72 ± 0.24 and 0.76 ± 0.24 μg, respectively. Females had higher IgG levels and total antibody levels compared to those of males. The difference between male and female IgG levels was statistically significant (*p*-value < 0.05).

**Figure 2 vaccines-10-01819-f002:**
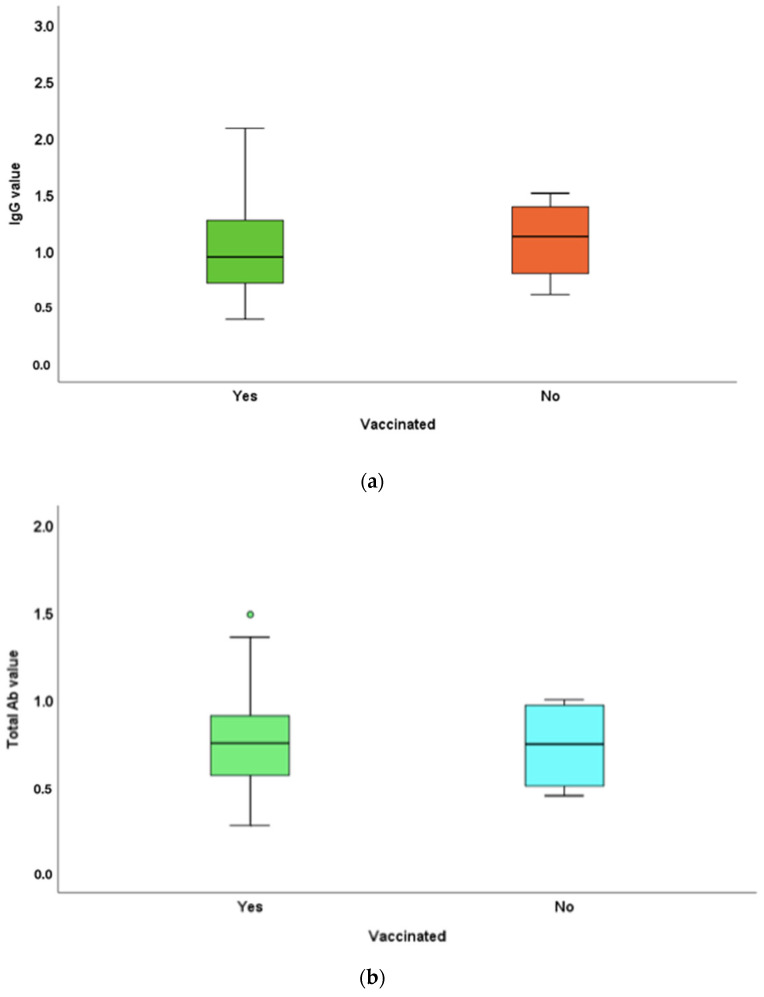
Mean IgG and total antibody titre differences between vaccinated and unvaccinated individuals in the saliva samples: (**a**) IgG titres in vaccinated and unvaccinated populations in the study were 1.02 ± 0.41 and 1.09 ± 0.39 μg, respectively. (**b**) Total antibody titres in vaccinated and unvaccinated populations in the study were 0.75 ± 0.24 and 0.73 ± 0.27 μg, respectively. IgG levels were slighter higher in the unvaccinated population, whereas total antibody levels were greater in vaccinated individuals. The differences in the antibody titres between the vaccinated and unvaccinated populations were not statistically significant (*p*-values = 0.65 and 0.92, respectively).

**Figure 3 vaccines-10-01819-f003:**
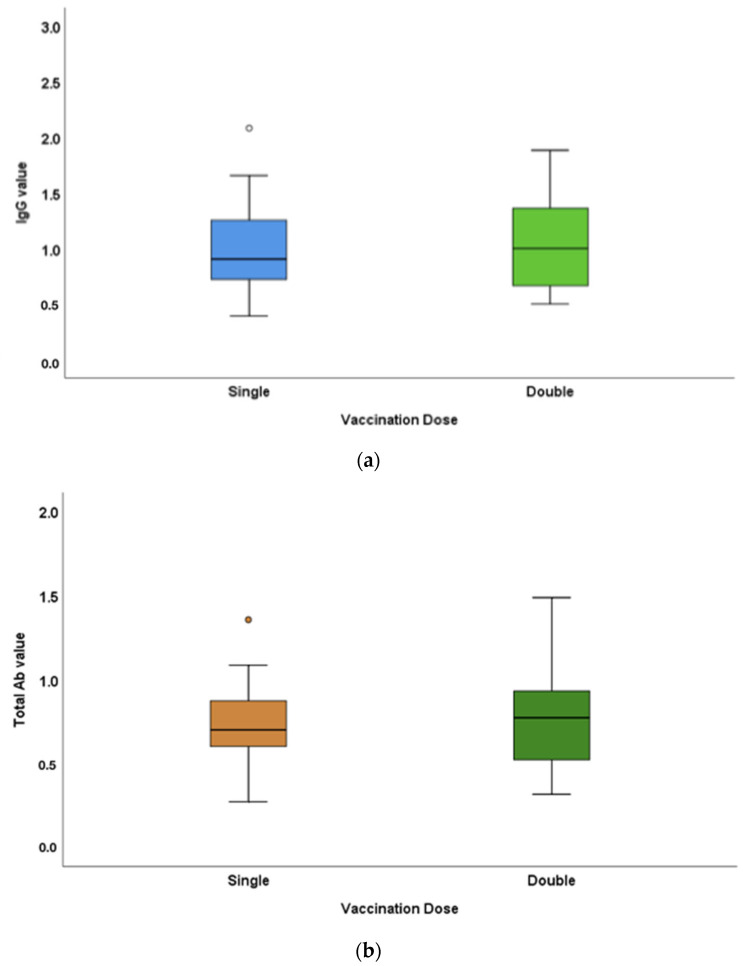
Mean IgG and Total antibody titre differences between single- and double-dose-vaccinated participants in the study: (**a**) IgG titres in single- and double-dose-vaccinated populations were 0.99 ± 0.37 and 1.06 ± 0.44 μg, respectively. (**b**) Total antibody titres in single- and double-dose-vaccinated populations in the study were 0.73 ± 0.22 and 0.76 ± 0.26 μg, respectively. IgG and total antibody levels were greater in double-dose-vaccinated individuals compared to those of the single-dose-vaccinated population. Differences in the antibody titres between the single- and double-dose-vaccinated population were not statistically significant (*p*-values = 0.75 and 0.65, respectively).

**Figure 4 vaccines-10-01819-f004:**
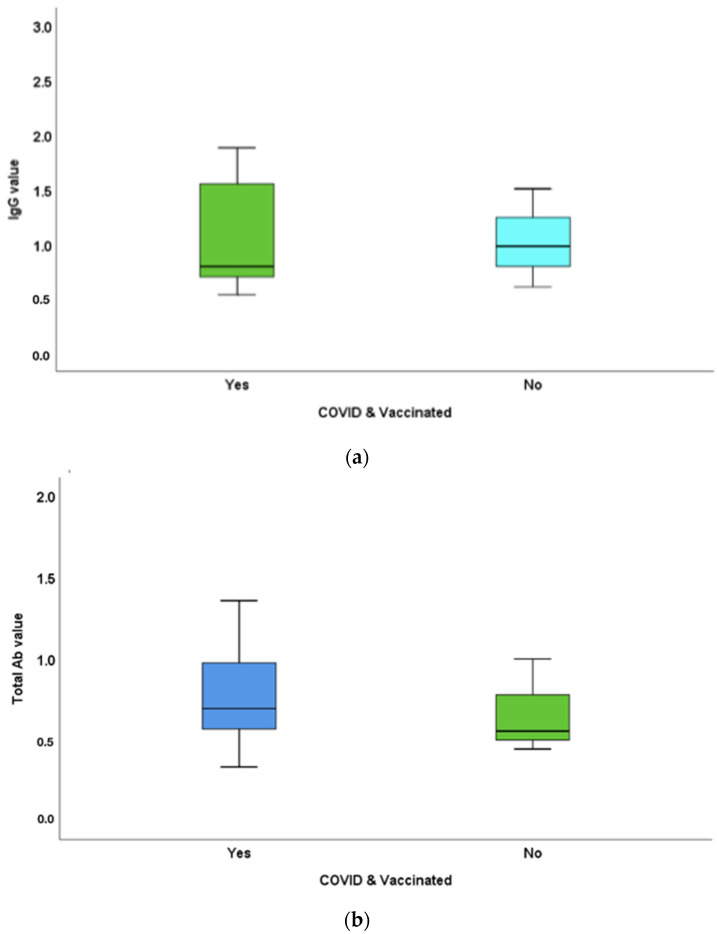
Differences between (**a**) mean IgG and (**b**) total antibody titres of study participants with a COVID-19 history with or without vaccination. (**a**) IgG in COVID-19-infected individuals with and without vaccination was 1.08 ± 0.49 and 1.03 ± 0.45 μg, respectively. (**b**) Total antibody titres in the COVID-19-infected population with and without vaccination were 0.78 ± 0.32 and 0.67 ± 0.29 μg, respectively. IgG and total antibody levels were in vaccinated individuals with COVID-19 history were greater compared to that of unvaccinated individuals with COVID-19 history. The IgG and total antibody titre differences in COVID-19 infected individuals with and without vaccination were not statistically significant (*p*-values = 0.81 and 0.48, respectively).

## Data Availability

The data presented in this study are available on request from the corresponding author.

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
