# Peer review of "Salivary Antibody Response of COVID-19 in Vaccinated and Unvaccinated Young Adult Populations"

_vaccines, 2022, doi:10.3390/vaccines10111819_

Round 1

Reviewer 1 Report

This research work is very original and current, the results obtained provide information on the performance of vaccines and their level of prevention against the SARS-CoV2 virus. Thanks to the authors for passing on their knowledge.

Author Response

Thank you for your insightful comments. Highly appreciated  

Reviewer 2 Report

The manuscript Salivary antibody response of Covid-19 in vaccinated and un- 2 vaccinated young adult population could provide information about antibodies response of COVID-19 in saliva samples. The topic of this work is appropriate for Vaccines, however, I listed a number of comments which I feel should be addressed.

Specific comments:

-Please update demographic data about COVID-19 in the abstract in introduction;

- To insert “Participants” in lower case. The work “subjects” could be more adequate.

-L20 – Based on English grammar “ as compared with males”.

-L25- Knowledge in lower case.

-L33. I suggest that you rephrase the sentence “ With many types of vaccines available, there is still no consensus about the effectiveness”. In my point of view, it is a consensus about the effectiveness of vaccines to COVID-19. It is a fact that vaccines against COVID-19 do not prevent the vaccinated population from being infected, especially a few months after vaccine administration, but advances and vaccine protection seem consensual in science. The opinion of a minority with anti-vaccine positions, often related to political aspects in some countries, should also not be generalized as a lack of consensus. Questions regarding the effect of the vaccine against COVID-19 and its variants can be addressed, but the description of effectiveness must be specific and not generalized.

-L37-43 . Please insert original studies as a reference in paragraph 2 of the introduction that performed serological analyzes in the population.

-In my point of view, the authors need improve the introduction with the current information about salivary SARS-CoV-2 antibodies.

-L50. The word “young” in the aim is not clearly. Perhaps, the sentence “young adult population (18-45 years)” used in material and methods is better. However, is not clear why the authors selected the population between 18-45 years. The motivation for choosing a population group must be clearly understood in the manuscript.

-L58 Change “covid infection, vaccination” for “COVID-19 infection and vaccination”.

- L65. To rinse mouth with water for 5 minutes is a longer time. Please confirm it.

-In the sample colletion, please inform the type of tube were the saliva was collected. Besides, insert a reference for “To eliminate debris, the saliva was immediately centrifuged 68 at 1800 rpm for 15 minutes at 4°C in a cooling centrifuge.”

-The material and methods section need to be improved in several parts with adequate references for each analysis and manufacturer of each reagent.

-In the results section , the word “nos” about the 25 or 65 subjects enrolled in the study is not clearly. 

-If 92% of the participants were in the age group of 18-21 years. Why you collected saliva from just 8% of your sample with 22-45 years old?

-Considering the importance of SARS-CoV-2 variants, the data of the collection need be inserted.

-Please, insert basic characteristics of both vaccines used in the study.

-L120 – Describe UAE.

- Considering that just 4.5%  or 4 subjects performed Group II as unvaccinated, the comparison with 85 subjects is very limited. The comparison about one or two doses is more appropriated (44 x 39 subjects)

- Please rephrase: “On consolidation of study participants information, Vac- 131 cinated individuals were not hospitalized, before vaccination it was very common.”

-L.139 - .93 and .43 … is it 0.93? 0.43? Improve it in the manuscript.

-In the abstract is informed that “The total antibody titer of vaccinated were raised than that of unvaccinated participants.”  However, in the results section, it stated that “the differences in both the antibody titers among the vaccinated and unvaccinated population were not statistically significant (p-value=0.65 and p-value=0.92).” Bearing in mind the 4 subjects in group II , the sentence in the abstract in wrong based on the results.

- I suggest that the number of samples for each comparison in the results section could be inserted. 

- The figures need to be inserted in the manuscript with the sentences about it. The section could be removed.

- In the Discussion section is critical offers proper references. The discussion about the effectiveness needs to be specific. In case of discussion about the effectiveness of vaccines could be important to inform if the vaccine was approved by WHO. It is important to understand about national or worldwide characteristics. 

- I suggest a general review in the discussion, it is necessary: 1) to point out limitations of this study for each comparison, as an example comparisons with groups of only 4 individuals. 2) In the no vaccine group, apparently 3 out of 4 individuals had COVID-19 and 1 did not, so this group seems inappropriate in this way. 3) the word Covid-19 is written in different ways, it is elegant a standardization. 4) A review of English is needed especially in the discussion. 5) Insert a clear comparison of this study, what is innovative or not with other studies? Please insert limitations of this study. 6) In comparison with other studies, it is important to compare the type of vaccine.

Author Response

We would like to thank the editor and reviewers for the comments and quick review. We believe that these changes will only improve the value of the manuscript All changes made in the manuscript have been carried out with track changes.

Reviewer 3 Report

1. Sample size difference between male and female is large might not give true indication that female have better immune response, the author mention it to many study that female have better immune response but that not fill the gap in the study.

2.The study should stress in types of vaccines given in title or in abstract because this study only tested two vaccines.

3.The author mention that IgG and saliva antibody have no significant effect on immune. but how about COVID-19 transmission is there any effect??

4. If the study included seniors or chronic diseases it will be stronger to test the increase in immunity 

Author Response

(The authors gave the same response as above.)

Reviewer 4 Report

I think this paper is no longer helpful for the readers after two years of pandemic. Moreover, the lack of a comparison (i.e. serum evaluation of antibody response ot antibody titer in serum in the same patients) did not allow any conclusion

Author Response

Thank you for your time and review. We strongly disagree with the reviewer’s comment –this study is the first of its kind in the young adults of the South Indian population. The necessity to compare with serum does not arise considering the evolution of salivary diagnostics.

Reviewer 5 Report

Sundar et al. attempted to assess the SARS-CoV-2 antibodies in saliva collected from vaccinated and unvaccinated participants using ELISA. They found that, as expected, the total antibody titer of vaccinated participants was raised than that of unvaccinated participants. That is, they succeeded in demonstrating the SARS-CoV-2 spike antibodies in saliva samples. They also showed the demerit of the use of serum samples, whereas they insisted the merit of the use of saliva samples as a noninvasive method. Before publication, you need to describe the details of the methods and results.

1. About 2.2.1 and 2.2.2, please describe the details of ELISA procedures. This is because we do not understand the relation between the optical density and the antibody titer as shown in unit.

2. Please check the expression of liter. Please use either L or l.

3. What is 60 l in Line 84?

4. About 3.1, please explain why you can express the titer in U/mL. Why can you convert the absorbance of ELISA to the titer?

Author Response

We would like to thank the editor and reviewers for the comments and quick review. We believe that these changes will only improve the value of the manuscript and all changes have been carried in the revised manuscript with track changes 

Round 2

Reviewer 3 Report

no comment